# Surface Characteristics of High Translucent Multilayered Dental Zirconia Related to Aging

**DOI:** 10.3390/ma15103606

**Published:** 2022-05-18

**Authors:** Flavia Roxana Toma, Mihaela Ionela Bîrdeanu, Ion-Dragoș Uțu, Roxana Diana Vasiliu, Lavinia Cristina Moleriu, Liliana Porojan

**Affiliations:** 1Department of Dental Prostheses Technology (Dental Technology), Center for Advanced Technologies in Dental Prosthodontics, Faculty of Dental Medicine, “Victor Babeș” University of Medicine and Pharmacy Timișoara, EftimieMurgu Sq. No. 2, 300041 Timisoara, Romania; flavia.toma@umft.ro (F.R.T.); roxana.vasiliu@umft.ro (R.D.V.); 2National Institute for Research and Development in Electrochemistry and Condensed Matter, 300569 Timisoara, Romania; mihaelabirdeanu@gmail.com; 3Department of Materials and Fabrication Engineering, Politehnica University Timişoara, Bd. Mihai Viteazu nr.1, 300222 Timisoara, Romania; dragos.utu@upt.ro; 4Department of Functional Science, “Victor Babeș” University of Medicine and Pharmacy Timișoara, EftimieMurgu Sq. No. 2, 300041 Timisoara, Romania; moleriu.lavinia@umft.ro

**Keywords:** multilayer zirconia, microroughness, surface topography, microhardness, aging

## Abstract

(1) Background: The purpose of this study was to evaluate the differences in terms of surface characteristics (roughness, topography, microhardness) among layers for multi-layered high translucent and super-high translucent zirconia and the influence of finishing and aging on surface characteristics and microstructure. (2) Methods: Three types of translucent multilayer zirconia were evaluated: STML (4Y-TZP); IPS e.maxZirCAD CEREC/in Lab MT Multi (4Y-TZP + 5Y-TZP); CeramillZolidfx ML (5Y-TZP). Ninety-six plate-shaped samples (32 for a material), 16 mm × 14 mm × 1 mm size, were cut with a precision disc, polished on both sides with sand papers and sintered respecting the manufacturer’s protocol. Half of the specimens (16) were finished by polishing and the other half by glazing and then equally divided into one control group and one group subject to aging by autoclaving (1 h, 134 °C, 0.2 MPa), resulting in four groups of eight samples, for each zirconia. The specimens were evaluated in three areas: cervical, medium, incisal-of each glazed or polished surface, before and after aging. Tests were performed to determine the surface roughness using a profilometer; the surface topography by an atomic force microscope (AFM) and a scanning electron microscope (SEM). Microhardness was recorded using a microhardness tester. Statistical analyses were performed using two-way ANOVA test, unpaired sample *t*-Test, paired sample *t*-Test (α = 0.05) and Pearson’s correlation. (3) Results: Before and after autoclaving, for glazed samples significance (*p* < 0.05) higher surface roughness, respectively lower microhardness in comparison with the polished group was assessed. No significant differences (*p* > 0.05) were reported between the three areas, on glazed or polished surfaces of a material. Although, after aging an increase in surface roughness was observed both on glazed and polished samples, statistical differences were found for STML (*p <* 0.05). No significant differences (*p*> 0.05) concerning microhardness among the same areas, on glazed and polished surfaces, recorded before and after aging, except CeramillZolidfx ML glazed samples. (4) Conclusions: For tested zirconia materials no significant differences among layers were registered regarding surface characteristics. Surface treatment (glazing or polishing) has a significant impact on surface roughness and microhardness. Both before and after aging, the surface roughness values for the glazed samples were higher than for those polished. The super translucent 4Y-TZP material was more affected by aging compared to the super-high translucent 5Y-TZP material. The combined material revealed similarities for each layer corresponding to the microstructure.

## 1. Introduction

To eliminate disadvantages as opacity, risk of chipping, substantial reduction of dental structure and also due aesthetics, biocompatibility and chemical stability compared to framework or conventional metal–ceramic, monolithic zirconia restorations are widely used in dentistry [1]. In order to obtain materials with high translucency, but also resistant, new grades of zirconia have been developed, pre-sintered blocks being available as pre-shaded in various tonalities or multilayered with gradients of chroma [2]. The translucent dental zirconia generations involved modifications of the crystal structure, increasing the content of Y₂O₃ (Yttrium) and cubic phase, resulting in two polycrystalline materials: 5Y-TZP and 4Y-TZP (partially stabilized zirconia with 5 mol% and respectively 4 mol% Yttrium) [3,4]. The main phase of zirconia is decisive for the individual characteristics of each zirconia system [5,6].

These compositional differences can influence three important clinical properties of the material: translucency, toughness, and resistance to hydrothermal aging [7].

The strategy for improving the optical performance of material imply microstructural variations—the higher Yttrium content influences the coefficient of thermal expansion, the size of grains (they are larger), reduce the residual porosity, resulting fewer boundaries, less birefringence, and scattering of the light [8], leading to higher translucency [6]. A recent study related that 5Yis 33%, and 4Yis 13% more translucent than 3Y-zirconia (3 mol% Yttrium stabilized-tetragonal zirconia polycrystal) [9]. The nanoparticles zirconia is presumed to be the next progress in evolution of this material [10,11,12].

In the microstructure, when stress occurs, the tetragonal phase ensures the ability to withstand compression, stopping the progression of the micro-cracks and elimination of microstructure defects [13], inducing transformation toughening. This mechanism is reduced for zirconia with higher cubic content. The super translucent 4Y-TZP (fourth-generation) contains 4 mol% Y_2_O_3_ and around 25% cubic phase. For super-high translucent 5Y-TZP (third-generation), the Y_2_O_3_ content was increased to 5 mol% and the quantity of the cubic phase rises up to 50%, also the amount of tetragonal phase decreases, leading to a lower possibility of tetragonal to monoclinic phase transformation, with reduction in strength and fracture toughness [8]; that is a disadvantage of these new translucent materials and the clinicians should use the 4Y and 5Y zirconia when the aesthetic needs are required.

Autoclave treatment induce an accelerate aging of zirconia [14,15].

Negative effects were reported after simulated accelerating aging of zirconia—LTD (low temperature degradation), by autoclaving [16], but limited types of materials were evaluated in short-term aging, hence different studies on multilayered translucent zirconia are essential for clinical applications. The aging process is associated with a slow tetragonal to monoclinic phase transformation, 4% volume increase of particles, micro-cracks with water penetration, grain detachment, decrease in hardness and fracture strength, surface roughening and degradation at the surface (and subsurface) which is in contact with moisture [1,17,18]. It was observed that hydrothermal and mechanical impact could cause unexpected phase transformation [19]. However, cubic zirconia undergoes almost no monoclinic transformation and hydrothermal degradation. Many studies revealed that after aging, 5Y-TZP was less affected than 3Y-TZP, but both experienced a strength reduction [20].

It was observed that the surface roughness and the microstructure characteristics can affect the strength of the material [21] and be relevant factors that influence the appearance, the final color, or bacterial accumulation [22,23].

The monolithic multilayer zirconia has polychromatic, pre-colored layers—enamel (incisal), transition 1 and 2 (medium), dentin-body (cervical)—simulating the shade and translucency of the natural teeth; with chroma and opacity increasing toward gingival area; the enamel layer looking the most transparent, while the dentin layer appears more opaque owingto the considerable low values for light permeability.

Several studies have reported that the first introduced multilayer zirconia (Katana, Kuraray Noritake), presented the same content of yttrium, respectively cubic phase in the layers of the material; among these being only the color difference given by the pigment composition [24]. Later, multilayer zirconia with different composition and microstructure in the same material (IPS e.maxZirCad Multi, IvoclarVivadent) was introduced.

The layers result as various formulations depending on the quantity of stabilizer and chemical composition. Because of these values, potential differences in the physical properties in the individual layers occur [6,25,26].

However, there are contradictory reports regarding the negative effects of the incorporation of oxide pigments on strength of zirconia [27,28] and by interface between layers was evidenced 30% lower strength than the bulk of material [29].

To improve aesthetics, polishing and glazing as surface treatments are applied [23,30]. It has often been shown that the polishing ability depends on microstructure of the material [31]. The slight polishing causes transformation nucleation usually around scratches due to tensile residual stress [32] and an increase in the monoclinic phase, may strengthen the material, but implies a directly proportional increase in surface roughness [33,34]. The gentle grinding and polishing using a small-grit (15–30µm) tool and a low-speed hand-piece, cause less-extensive tetragonal to monoclinic transformation and induce minor or no obvious flaws [35,36,37] in zirconia with different Yttrium concentrations [38].

Glazing as a laboratory procedure is achieved through applying a blend of colorless glass powder and liquid layer that reduced the roughness, seals the pores, and smoothensthe ceramic surface [39,40]. The appearance of the glazed surface can be influenced by the layer deposition technique, the special brush, layer thickness, and uniformity of zirconia interface features, sometimes resulting in an irregular wavy surface with small defects. According to various results, after aging, the glaze layer may be negatively affected, broken, or separated from thematerial [41,42].

It is known, when the dopant concentration is high, the surface roughness is increased (meaning for 8Y-or 5Y-TZP was higher in comparison to 3Y-TZP) [43,44]; also, it causesan increase incubic phase amount and lowers the strength and fracture toughness, [1,6] and lowers the mechanical properties [45]. As a result, the surface treatment as glazing may or not have a protective role; polishing may induce defects and/or phase transformation which can lead to an increase or decrease in strength of the material [32,46,47,48].

Glazing and polishing enhance the appearance of restorations, decrease the bacterial accumulation, the inflammatory tissue response and the wear of opposing teeth [31,32,33].

This in vitro study aims to evaluate: (a) The differences in terms of surface characteristics (roughness, topography, micro-hardness) among layers for multi-layered high translucent and super-high translucent zirconia; (b) how glazing and polishing affect the surface properties of the mentioned materials; (c) the influence of LTD (low temperature degradation) as aging method, on surface characteristics and microstructure.

The null hypotheses were: (1) There are not significant differences regarding the surface texture and microhardness in the three areas (cervical, medium, incisal) of the materials; (2) glazing and polishing affect the surface properties of the material; (3) the roughness value increases and microhardness decreases after the aging procedure.

## 2. Materials and Methods

Three types of commercially multilayered dental zirconia with various translucencies and Yttrium content wasinvestigated in this study: super-translucent zirconia STML (Katana, Kuraray Noritake Dental, Tokio, Japan)-noted ST; IPS e.maxZirCAD CEREC/in Lab MT Multi (IvoclarVivadent AG, Schaan, Liechtenstein)-noted IP and super-high-translucent zirconia CeramillZolidfx ML (Amman Girrbach, AG, Koblach, Austria)-noted CE. The properties of these materials are presented in Table 1.

### 2.1. Specimen Preparation

Ninety-six specimens (32 samples of each zirconia) were cut from presintered blocks of each material, with a precision disc, followed by polishing on both sides using #600, #1500, #2000 sandpapers. Each material was sintered in a ceramic furnace LHT 01/17D (Nabertherm, Lilienthal, Germany) 1550 °C/7h-for STML; DekemaAustromat 674 (DEKEMA Dental-Keramiköfen GmbH, Freilassing, Germany) 1500 °C/9 h 50 min-for IPS e.maxZirCAD CEREC/in Lab MT Multi; Programat S2 (IvoclarVivadent, Schaan, Liechtenstein) 1450 °C/8h-for CeramillZolidfx ML according to a protocol recommended by each manufacturer. Half of the specimens (16 samples) were finished only by polishing and the other half by glazing. The finishing procedure was performed on one surface of the plate. Further, each assortment was divided equally into one control group and one group subject to aging by autoclaving (1 h, 134 °C, 0.2 MPa), as LTD (low temperature degradation). Four groups (n = 8) resulted for each material: control glazed (cg), control polished (cp), autoclaved glazed (ag), autoclaved polished (ap). The glazed (g) and the polished (*p*) surfaces were evaluated into three areas corresponding to the layers: cervical (c), medium (m), incisal (i), before, and after aging.

Glazing was performed for all specimens with two thin layers—IPS Ivocolor Glaze Paste (IvoclarVivadent, Schaan, Liechtenstein) in Vita Vacumat 6000 M furnace (Bad Sackingen Germany), in accordance with the manufacturer’s indications. For polishing a low-speed device, polishing brush and diamond paste-Renfert Polish (Renfert, Hilzingen, Germany) was used, following the same technique to obtain a smooth and glossy surface. The final dimensions of plate-shaped samples were 16 mm × 14 mm and 1 mm thickness, andweremeasured with a caliper. For a better understanding a flowchart was created, inwhichthe studied groups are described-Figure 1.

### 2.2. Hydrothermal Aging

The samples were subjected to the artificial aging process as LTD (low temperature degradation) using an autoclave (Sterilclave 24 B, Cominox, CarateBrianza, Italy) with distilled water for 1 h, at 134 °C and 0.2 MPa; it is known that 1 h of autoclaving in these conditions would induce on zirconia, the same effects as 3–4 years in vivo [14].

### 2.3. Surface Roughness and Topography

The measurement of surface roughness was made using a contact profilometerSurftest SJ-201(Mitutoyo, Kawasaki, Japan) with a 2 µm diamond stylus. On each level (cervical, medium, incisal), three measurements were recorded in three randomly selected areas, on polished and on the glazed samples, respectively. The values of the Ra (µm) and Rz (µm) parameters were obtained. Ra (µm) is arithmetic average surface roughness and Rz (µm) is maximum surface roughness. The cut-off value was 0.3 mm to a force applied by 0.7 mN.

### 2.4. Atomic Force Microscopy (AFM)

The nanosurface characterization of the specimens was evaluated with an atomic force microscope Nanosurf Easy Scan 2 Advanced Research (Nanosurf AG, Liestal, Switzerland). The values of nanoroughness parameters Sa (nm)-arithmetical mean height and Sq (nm)-root mean square height resulted. For surface topographic analysis, AFM provided three-dimensional images that were achieved with a scan size of 2.2 × 2.2 µm, in non-contact mode.

### 2.5. Scanning Electron Microscopy (SEM)

To investigate the surface modification resulting from different surface treatments, a scanning electron microscope with high resolution field emission—Inspect S (FEI Company, Hillsboro, OR, USA) was used. The assessment of surface characteristics was performed on both samples (glazed, polished), in the three individual colored areas (cervical, medium, incisal).

### 2.6. Microhardness Testing 

A microhardness tester (HVS-10A1, Huatec, Bejing, China), was used to assess the surface hardness of glazed, respectively polished surfaces of the samples, on each domain (cervical, medial, incisal) in three randomly selected areas. The indentation technique determined the Vickers Hardness number (VN), by applying a force of 0.3 kg for 10 s.

### 2.7. Statistical Analysis 

Two different programs were used for statistical analyses—the IBM SPSS Statistics software (IBM, New York, NY, USA) and JASP (v.16.2, University of Amsterdam, Holland). In the first part was used descriptive statistics, calculating the central tendency and dispersion parameters. To decide upon the statistical type of tests we applied the Shapiro–Wilk test for distribution and obtained that the data are normally distributed (*p*> 0.05) in most of the cases; for a complete perspective, parametrical and non-parametrical tests were applied. The repeated measures ANOVA test was used for statistical analysis of more than two-time moments/dependent groups. The unpaired *t*-Test was applied for two different groups (for statistical analysis of variables recorded on glazed and polished sides); the paired *t*-Test was performed for two-time moments situation (for comparing areas of treated surfaces-before and after aging, for a material and the significant differences because of autoclaving).

A correlation model between Ra and microhardness, was performed—the straightness of the correlation was measured using Pearson’s coefficient. For the entire study, a significance level of α =0.05 was established.

## 3. Results

### 3.1. Micro-Surface Roughness

Mean values and SD (standard deviation) for Ra (µm), arithmetic average surface roughness and Rz (µm), maximum surface roughness measurements on glazed and polished samples, before aging are summarized in Table 2.

Among glazed tested groups, the lowest mean Ra values were recorded for ST (0.064 to 0.067) and the highest for CE (0.086 to 0.084), with very close values between layers. For IP, the values tend to increase from the cervical (0.073 ± 0.009) to incisal (0.083± 0.009) area. 

Among the polished tested groups, the lowest Ra values were measured for ST (0.041 to 0.043) and the highest for CE (0.045 to 0.047), which were almost similar in all three areas. For IP, an upward trend was registered, the lowest value (0.043 ± 0.010) was in the cervical area and the highest (0.046 ± 0.009) in the incisal one.

Mean values and SD (standard deviation) for Ra (µm)-arithmetic average surface roughness and Rz (µm)-maximum surface roughness measurements after aging are summarized in Table 3.

Among glazed tested groups, it was observed that the surface roughness increased in all areas for the three materials, the lowest mean Ra values were recorded for ST (0.077 to 0.080) and the highest for CE (0.086 to 0.088), with very close values between layers. For IP, the Ra values increased significantly in the cervical area and slightly in incisal area; however, the surface roughness is lower in cervical area (0.081 ± 0.009) than in incisal area (0.086 ± 0.010) for this material.

Among polished tested groups the lowest Ra values were measured for ST (0.052 to 0.054) and the highest for CE (0.047 to 0.048), almost similar in all three areas. For IP, the values tend to decrease from the cervical (0.053 ± 0.009) to incisal (0.049 ± 0.009) area.

For the glazed side, the lowest Rz values was recorded for ST (0.512 to 0.435) and the highest (0.576 to 0.591) for CE and for IP an intermediate appearance with the lowest value (0.534 ± 0.094) in cervical area and the highest (0.562 ± 0.089) in incisal ones.

On the polished side, the lowest Rz values (0.406 to 0.416) were reported for CE and the highest (0.518 to 0.535) for ST.

Mean roughness measurements performed before and after aging, in three areas (cervical, medium, incisal) on the glazed and polished samples (n = 8) are presented in Figure 2.

Before aging, the two-way ANOVA test (α = 0.05) reported insignificant differences (*p >* 0.05) when the statistical test was performed to compare the surface roughness between cervical-medium-incisal areas individual on glazed (for Ra: ST *p* = 0.771, IP *p* = 0.812, CE *p* = 0.909; for Rz: ST *p* = 0.504, IP *p* = 0.587, CE *p* = 0.704), respectively on polished samples (for Ra: ST *p* = 0.814, IP *p* = 0.692, CE *p* = 742; for Rz: ST *p* = 0.784, IP *p* = 0.831, CE *p* = 0.863); also after aging, no significant differences (*p >* 0.05, α = 0.05) were observed when comparing the every three areas, on each glazed or polished surface, for a material.

Both before and after aging, the surface roughness values for the glazed samples are higher than for those polished. The unpaired *t*-Test test (α = 0.05) reported a significant difference (*p <* 0.05) in terms of Ra-arithmetic average surface roughness and Rz-arithmetic average surface roughness betweenthe glazed and polished groups in the same areas, for all three materials; the *p* values are included in Table 4.

Although it has been observed that the surface roughness has increased on polished, respectively on glazed side after hydrothermal aging, the statistical paired *t*-Test (α = 0.05) reported significant differences concerning Ra, Rz among the samples before—after autoclaving only for ST, the *p* values are included in Table 5.

### 3.2. Atomic Force Microscopy (AFM)

Before aging, for each kind of material, fine and deeper interconnecting striations and scratches, small irregularities as scattered and sharp peaks were observed on both treated surfaces in all three areas. Among materials, for CE, the surface irregularities are more obvious both in the group of glazed and those of polished samples.

The values for parameters Sa (nm)-arithmetical mean height and Sq (nm)-root mean square height resulted for the glazed and polished samples are showed in Table 6.

The AFM 3D images of the control group recorded on each area of glazed samples are shown in Figure 3.

The AFM 3D images on each area of the polished samples of the control group are shown in Figure 4.

After aging, the values for Sa (nm)-arithmetical mean height and Sq (nm)-root mean square height resulted for the glazed samples and polished samples are shown in Table 7.

Sa = arithmetical mean height, Sq = root mean square height. ST = STML, IP= IPS e.maxZirCAD CEREC/in Lab MT Multi, CE = CeramillZolidfx ML. ag = aged-glazed, ap = aged-polished, c = cervical, m = medium, i = incisal.

After aging, the AFM 3D images recorded the topography on each area of the glazed samples—Figure 5.

After autoclaving, the AFM 3D images on each area for polished samples are represented in Figure 6.

After aging, there is a significant increase in Sa (nm)-arithmetical mean height and Sq (nm)-root mean square height values, for all materials, both on the glazed and polished surfaces.

On the glazed samples Sa (nm) values have an ascending aspect from ST to CE. ST in the medium area recorded the lowest value 21.41 ± 0.02 than 2.42 ± 0.01-before; the highest value resulted for CE in cervical area 31.81 ± 0.01, compared to 1.72 ± 0.01-before. For IP in medium area was reported an intermediate value 29.05 ± 0.02 compared to 2.54 ± 0.01-before.

On the polished samples, the lowest value was reported for CE in cervical area 20.34 ± 0.01, compared to 1.39 ± 0.01, before and the highest for ST in incisal and medium area 31.78 ± 0.02 than 1.32 ± 0.02, before. For IP, the values have an intermediate aspect, but they are almost similar to those on glazed side.

On glazed side of specimens, the lowest Sq value is in medium area for ST-24.41 ± 0.02 compared to 2.94 ± 0.01, before and the highest for CE in incisal area 36.37 ± 0.01, compared to 1.98 ± 0.01, before. For IP in medium area Sq value 33.22 ± 0.02 was recorded, compared to 3.05 ± 0.01, before.

On the polished side of samples, the lowest Sq values werereported for CE in cervical area 23.52 ± 0.01, compared to 1.74 ± 0.01, before and the highest for ST in medium area 36.25 ± 0.02 than 1.69 ± 0.02, before.

Before aging, the two-way ANOVA test (α = 0.05) reported no significant difference (*p >* 0.05) regarding to Sa = root mean square height, Sq = root mean square height, between the three areas (cervical-medium-incisal) of glazed or polished samples for a material.

The unpaired *t*-Test test (α = 0.05) reported significant differences (*p <* 0.05) in terms of Sa-arithmetical mean height in cervical and medium areas and no significant differences in incisal area between glazed and polished surfaces of each material (incisal: ST glazed-polished *p* = 0.08; IP glazed-polished *p* = 0.241; CE glazed-polished *p* = 0.452).

After aging, the two-way ANOVA test (α = 0.05) reported no significant difference (*p >* 0.05) regarding to Sa = root mean square height, Sq = root mean square height, between the three areas(cervical, medium, incisal) for each glazed or polished samples, of a material and the unpaired *t*-Test test (α = 0.05) reported significant differences (*p <* 0.05) in terms of Sa-arithmetical mean height, between the same areas of the glazed and polished surfaces of each material, for ST in cervical and medium area.

The before–after aging statistical paired *T*-Test (α = 0.05) found significant differences (*p* ≤ 0.05) regarding Sa, Sq values among the same areas of glazed, respectively polished samples.

### 3.3. Scanning Electron Microscopy (SEM)

The surface topographic analysis revealed a regular and homogeneous structure for the glazed samples, before aging, except some places with a wavy surface due to the application of glaze. In the medium area on ST followed by IP samples, the surface displayed interconnecting striations and was rougher than the other areas; by IPS and CE in incisal area, fine striations are visible under glaze layer.

On the polished samples, before aging, the striations, small irregularities, and defects are more obvious than on the glazed surfaces; for ST and IP in medium area, flaws and porosities were observed on the surface.

SEM images of the control samples are illustrated in Figure 7 and Figure 8.

After aging for glazed specimens, the appearance of the surfaces is quite similar; however, on ST samples in cervical and medium area, striations and micro-cracks may be observed. After autoclaving, for CE, the appearance of the surface is quite similar to that before aging. On polished samples, there was an obvious damage in all areas compared to the situation before aging.

After aging, the SEM images of samples are illustrated in Figure 9 and Figure 10.

### 3.4. Microhardness Testing

Before and after aging, the mean Vickers hardness values with SD (standard deviation) of the materials, for both glazed and polished specimens in three areas are presented in Table 8.

On the glazed samples, the highest microhardness values were reported for ST (770.33 ± 9.60), followed by IP (700 ± 5.72) and CE (632.67 ± 8.50) for all, in the cervical area. On the polished samples, the highest surface microhardness was recorded for ST in cervical area and medium area (1763 ± 17.02), followed by IP in cervical area (1732 ± 15.70) and CE in cervical area (1637 ± 12.50).

Among materials, for CE, on the glazed and polished surface, the microhardness values were lower.

After aging, the microhardness for polished samples issignificantly higher compared to the glazed ones.

On the glazed side, for ST, the values decreased, the highest was recorded (672.67 ± 8.32) in cervical area; for IP the values are almost the same as before aging (ex. 694.33 ± 4.21) in incisal area. For CE, an increase in all three areaswas reported; the highest values (722 ± 8.11) were recorded in medium and incisal area.

On the polished samples, a minor increase in values can be observed; the highest (1667 ± 5.70) was reported for CE in incisal area.

The microhardness values recorded before and after aging on glazed and polished samples, in cervical, medium, incisal areas, for all three materials (n = 8), are summarized in Figure 11.

Before aging, the microhardness values for polished specimens were significantly higher compared to the glazed ones.

There were insignificant differences (*p >* 0.05) when the statistical test two-way ANOVA was performed to compare the surface microhardness between cervical-medium-incisal areas individual on glazed and on polished sides of each zirconia.

The unpaired *t*-Test test (α = 0.05) assessed a significant impact (*p <* 0.005) to the analysis in the same areas between control-glazed and control-polished groups, for all three materials.

For both glazed and polished surfaces, Pearson’s correlation reported a negative correlation between surface roughness Ra and microhardness, r = −0.9, which means that the hardness decreases when the roughness is increased.

After autoclaving, no significant differences (*p >* 0.05) were observed by statistical two-way ANOVA test between every two or three areas, on each glazed or polished surface for a material.

The unpaired *t*-Test test (α = 0.05) revealed significant differences (*p <* 0.005) to the analysis in the same areas between aged-glazed and aged-polished groups, for all three materials.

The statistical paired *t*-Test (α = 0.05) reported no significant differences concerning microhardness values (Vickers Hardness number-VN) among the same areas recorded before and after aging, on the glazed and polished surfaces, except CE glazed samples; the *p* values are included in Table 9.

## 4. Discussion

The study investigated the surface characteristics, microstructure and microhardness on glazed and polished samples of three multilayered dental zirconia with different percentages of Y_2_O_3_ (4 mol%; 4 mol%+ 5 mol%; 5 mol%), first before and then after aging, achieved through autoclaving, as LTD (low temperature degradation). 

**Before aging**, among materials ST = STML, IP = IPS e.maxZir CAD CEREC/in Lab MT Multi, CE = CeramillZolidfx ML for the glazed samples, the lowest Ra-arithmetic average surface roughness values were found for ST (4 mol% Y_2_O_3_), followed by IP (4 mol% + 5 mol% Y_2_O_3_) and the highest surface roughness was recorded for CE (5 mol% Y_2_O_3_)-the superhigh-translucency zirconia, with the highest Yttrium content and the lowest flexural strength (700 MPa) of the three analyzed materials.

Among areas of the glazed samples, for ST, the highest Ra value (0.067 ± 0.010) was recorded in the incisal area; for IP, it was found that surface roughness increases from cervical (0.073 ± 0.009) to incisal (0.083 ± 0.009); and for CE close values were reported, the highest cervical (0.086 ± 0.010). Between the glazed (cervical-medium-incisal) areas of a material no statistically significant differences were found (*p <* 0.05). On the polished samples, the same ratio was observed as for glazed specimens, the lowest Ra values were found for ST (ex. 0.043 ± 0.010—medium), followed by IP (ex. 0.046 ± 0.009-incisal) and the highest for CE (ex.0.047 ± 0.008-cervical). Among (cervical-medium-incisal) areas on glazed or polished surfaces of a material no statistically significant differences (two-way ANOVA test, α = 0.05) were found (*p <* 0.05).

The surface roughness values measured for the glazed samples are higher than for those polished. The unpaired *t*-Test (α = 0.05) reported a significant difference (*p <* 0.05) in terms of Ra and Rz between the same areas of glazed and polished sample, for all three materials.

When comparing the surface roughness obtained between the areas of a material, it is observed that for ST, respectively for CE, the values are close to each other, both on glazed and polished samples; which means as some manufactures and some studies show, the layers presented almost the same content of Yttrium, cubic phase, respectively a similar microstructure; the color difference being given only by the pigment composition [24]. Instead, for IP, the surface roughness values reported an ascending orderfrom cervical to incisal, corresponding to the various distribution of Yttrium content and cubic phase, respectively, and a different microstructure among the layers of this material.

Regarding glazing process as a surface treatment, the microstructure, the smoothness of the surface on which the glaze layer is deposited, and the interface between them greatly influence the external appearance of the surface. For ST, which contains smaller grains than CE, the surface roughness was lower. It is also possible to record different Ra, Rz values when using the manufacturer’s own glaze pasta.

Referring to the polishing process, for this study small-grit grinding tools were used so as not to cause, as much as possible, phase transformation and surface degradation which results in an increase in surface roughness. However, a zirconia such as CE, that has the lowest mechanical properties among the studied materials and contains larger grains that can be easily dislocated during processing resulting unevenness and flaws, presented the highest surface roughness.

In addition, for a better surface topography evaluation, the samples were examined using an atomic force microscope (AFM) and a scanning electron microscope (SEM).

AFM provided a high resolution three-dimensional visualization of glazed and polishing samples, the values for parameters Sa (nm)-arithmetical mean height and Sq (nm)-root mean square height resulted. The nanoroughness values found on the glazed surfaces were higher than on the polished ones. On the glazed samples, the highest Sa value was recorded for CE (2.66 ± 0.02), followed by IP (2.54 ± 0.01) and the least affected was ST (2.42 ± 0.01), all in the medium area. On the polished surfaces, the highest Sa (nm) value resulted for CE (1.58 ± 0.01) in medium area, IP (1.55 ± 0.01), and almost the same for ST (1.53 ± 0.02) in incisal area.

The SEM examination for glazed samples revealed a fairly smooth and uniform surface with striated patterns resulting from the deposition (brushing) of the glaze layers, creating a wavy appearance, which might influence the surface roughness. By CE and IP-incisal areas, the striations resulting from processing are visible under the glaze layers. The images which present the polished surfaces showed interconnecting scratches, shallow striation and micro-cracks created by the rotative instrument on a smooth surface, except for ST and IP medium area, where grater damage was observed.

Surface microhardness test measures the resistance of material to successive indentation. Mechanical properties are influenced by the crystalline phase, yttrium content, more precisely by the microstructure of the material [52]. In this study, the unpaired *t*-Test test (α = 0.05) reported a significant difference (*p <* 0.005) regarding the microhardness (VN) between polished (1552 to 1763 MPa) and glazed surfaces (597 to 763 MPa).

The glaze layer was locally negatively damaged; thus confirming that the surface treatment can affect the surface hardness and the resistance of the material. The sharpest and most chipped edge of the indent was observed on the glazed surface of CE (5 mol% Y_2_O_3_), followed by ST (4 mol% Y_2_O_3_) and the least obvious crack patterns and indentation imprints by IP (4 mol% + 5 mol% Y_2_O_3_).

Considering that the same glaze was used for all three materials, it is possible that the microstructure, surface appearance, the interface had an influence on the sealing, adhesion capacity, and resistance of the glaze layer. Moreover, it may be relevant as the glaze (IPS Ivocolor Glaze Paste) and IP (IPS e.maxZirCAD CEREC/in Lab MT Multi) materials were produced by the same manufacturer (IvoclarVivadent, Schaan, Liechtenstein). Among the three areas of glazed, respectively of polished surfaces, no significant differences (two-way ANOVA test, *p >* 0.05) were reported, confirming the first hypothesis, also in terms of hardness.

For both glazed and polished specimens, a negative Pearson’s correlation (r = −0.9) between surface roughness and microroughness was reported; meaning that increasing the roughness implies decreasing the hardness and the surface treatment significantly influenced the surface roughness, respectively the microhardness [53].

**After aging**, both glazed and polished samples show an increase in roughness and the values recorded on the glazed samples were higher than for those polished ones.

Among glazed specimens, compared with the situation before aging, statistical paired *t*-Test (α =0.05) revealed a significant increase in values for ST (*p <* 0.05) and an insignificant increase for CE (*p >* 0.05); however, the lowest mean Ra values were recorded for ST (0.077 to 0.080) and the highest for CE (0.086 to 0.088). For IP, the surface roughness increased in cervical area and slightly in incisal area, but the values were lower in cervical area (0.081 ± 0.009) than in incisal area (0.086 ± 0.010).

As a result, the conclusion could be that the material with 4 mol% Yttrium content (ST, IP–cervical) undergoes a more intense aging process accompanied by phase transformation, volume increase of particles, grain detachment, and surface roughness. For CE (5 mol% Yttrium), the surface roughness before aging was the highest and after aging showed a minor increase, meaning that the superhigh-translucency zirconia is not very affected by aging; however, it remains the material with the roughest surface.

Other studies demonstrated as well that the increased roughness does not accelerate the aging process for (super) high-translucent zirconia and the toughness was preserved after grinding and LTD [16,54].

Certainly, the glaze layer has a protective role, but may be directly affected by the aging process or as a result of alteration in the underlying material, with influence on surface texture and hardness.

Among polished samples, compared with the situation before aging, the paired *t*-Test (α = 0.05) reported that the surface roughness values for ST (*p <* 0.005) increased more than those for CE (*p >* 0.005) and among materials, the highest Ra values were measured for ST (0.052 to 0.054) and the lowest for CE (0.047 to 0.048); for IP, the values tend to decrease from the cervical (0.053 ± 0.009) to incisal (0.049 ± 0.009) area. This means that the material with 4 mol% Yttrium content is more susceptible to the aging process, thus hydrothermal and mechanical impact cause phase transformation, resulting in surface degradation and an increase in microhardness; this situation has been presented in other studies [55,56].

On the glazed surfaces, the nanoroughness values Sa = arithmetical mean height, Sq = root mean square height, were higher than on the polished ones;the lowest values were found for ST and the highest for CE. On the polished specimens, the highest nanoroughness values were recorded for ST and the lowest for CE. Regarding Sa and Sq, the paired *t*-Test (α = 0.05) reported a significant increase in glazed and polished specimens, after aging.

After aging, the surface topographic analysis (SEM) revealed for ST glazed specimens, more visible striations, and micro-cracks (cervical, medium- area), compared to the appearance before aging. For CE, a minor increase in surface roughness was recorded and the surface aspect is quite similar to that before aging. On polished samples compared to the initial situation, in all areas, an evident degradation with deeper striations and scratches, sharp peaks, irregularities and defectswas observed.

It was revealed that both before and after autoclaving, glazing and polishing as surface treatment, influenced and affected the microroughness and topography of zirconia with different Yttrium content. The second null hypothesis is accepted.

Although an optimal threshold has not been set, it was reported that the roughness value (Ra-arithmetic average surface roughness) above 0.2 µm, could induce negative effects such as retention of bacteria, periodontal inflammation, dental caries risk [57], and a value above 0.3 µm can be perceived by the tongue, negatively affecting patient comfort [57,58]. The Ra values results of this study, for all three zirconia types, in three areas on glazed, respectively polished surfaces, were lower than 0.2 µm, therefore they may be considered clinically acceptable.

Regarding microhardness, after autoclaving, the unpaired *t*-Test test (α = 0.05) revealed that the values for polished samples are maintained significantly higher (*p <* 0.005) compared to the glazed one, confirming the second hypothesis, also in terms of hardness.

On the glazed specimens, for ST was observed a decrease in microhardness, while those for IP remained at a level similar to that before aging and CE, denotingan increase in values. On the polished samples, for ST and IP a minor increase of values was recorded and for CE the values remain approximately the same as those before aging.

Before aging, ST (4 mol% Yttrium) recorded the lowest surface roughness and the highest microhardness, demonstratingthe best mechanical properties among the studied materials, on the glazed and polished samples. After aging, this material experienced the largest increase in roughness for both groups and microhardness decreased on glazed and increased on polished side. It seems that for the glazed samples, the aging process induced t-to-m phase transformation with an increase in surface roughness, adversely affecting the interface features and the glaze layer and for the polished specimens, the phase transformation led to a toughening reaction.

For CE (5 mol% yttrium), before autoclaving, the microhardness values were close to those for IP, but lower than those measured for ST on glazed surfaces and the lowest among materials on polished samples. After aging, the microhardness increased for the glazed group and between materials was the highest; for the polished samples the values are maintained (with a slight increase) as before aging and among materials were the lowest. It seems that the material was not very affected by aging process, however the hardness increased and the glaze layer had a protective role.

For IP (4 mol% + 5 mol%), regarding mechanical properties an intermediate aspect among the tested materials, both on glazed and polished samples, before or after autoclavingwas observed. The records revealed similarities between properties and values measured in cervical area with those of ST and the incisal area with those of CE.

The aging behavior is correlated with the microstructure of the material, and with the surface treatment. As a result of these findings, the third null hypotheses—the roughness value would increase and microhardness decrease after the aging procedure—is not accepted.

The limitation of this study was the short-term aging and the fact that certain factors that interact with restoration in oral environment could not be fully reproduced (chewing force, saliva components, temperature, and pH fluctuations).

## 5. Conclusions

For tested zirconia materials, no significant differences among layers were registered regarding surface characteristics.Surface treatment (glazing or polishing) has a significant impact on surface roughness and microhardness; the glazed samples were found with higher surface roughness and lower microhardness compared to the polished ones. This correlation is preserved even after LTD.After aging microroughness increases, but significant only for the super translucent material. There is a significant increase in nanoroughness values, for all materials, both on the glazed and polished surfaces. Related to microhardness, aging does not have a significant influence.The super translucent 4Y-TZP (fourth-generation) material was more affected by aging compared to the super-high translucent 5Y-TZP (third-generation) material. The material with 4 mol% + 5 mol% revealed similarities for each layer corresponding to the microstructure.

## Figures and Tables

**Figure 1 materials-15-03606-f001:**
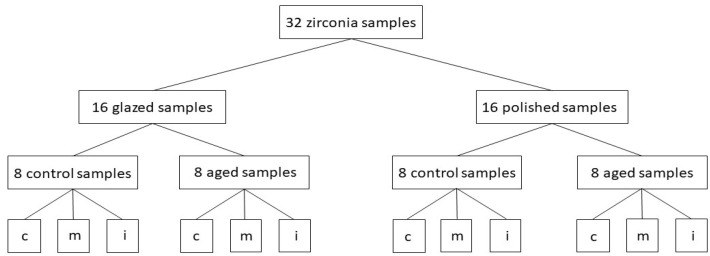
The flowchart of the study in which the main samples and groups were plotted and the way they are divided; c = cervical area, m = medium area, i = incisal area.

**Figure 2 materials-15-03606-f002:**
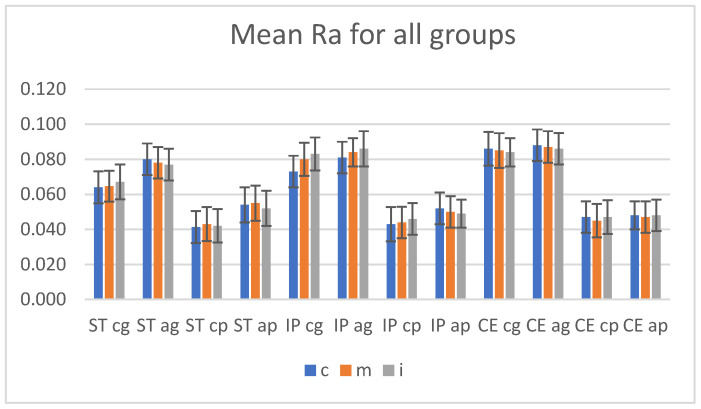
Mean Ra values (arithmetic average surface roughness) with SD (standard deviation), before and after aging for all groups. ST = STML, IP = IPS e.maxZirCAD CEREC/in Lab MT Multi, CE = CeramillZolidfx ML. cg = control glazed, cp = control polished, ag = aged glazed, ap = aged polished, c = cervical, m = medium, i = incisal.

**Figure 3 materials-15-03606-f003:**
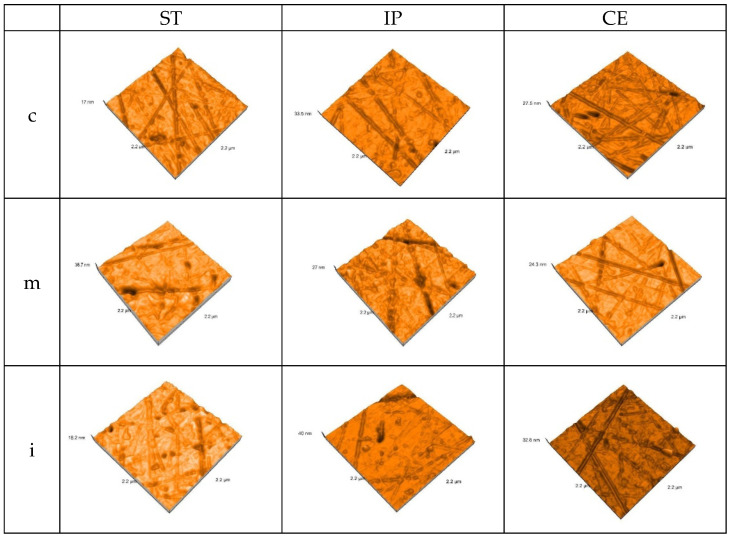
3D AFM images of the glazed samples, before aging.ST = STML, IP = IPS e.maxZirCAD CEREC/in Lab MT Multi, CE = CeramillZolidfx ML. c = cervical, m = medium, i = incisal.

**Figure 4 materials-15-03606-f004:**
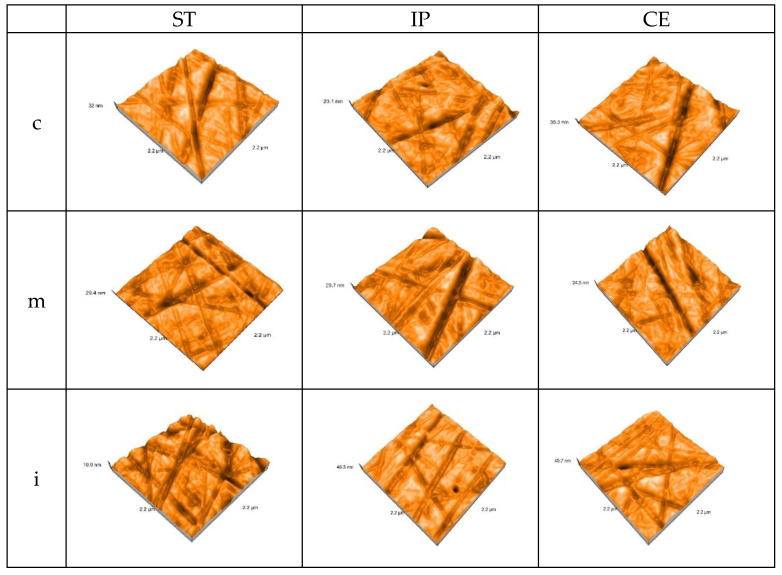
3D AFM image of the polished samples of the control group. ST= STML, IP= IPS e.maxZirCAD CEREC/in Lab MT Multi, CE= CeramillZolidfx ML, c = cervical, m = medium, i = incisal.

**Figure 5 materials-15-03606-f005:**
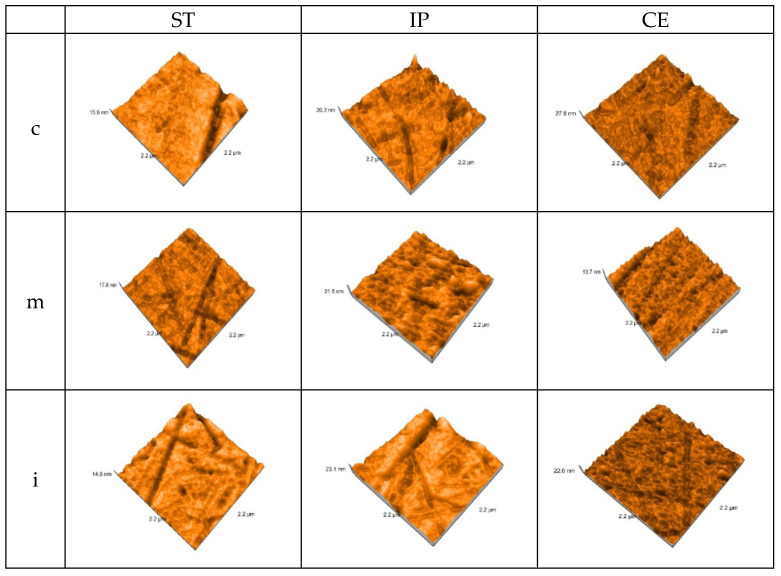
3D AFM images of the glazed samples, after aging.ST = STML, IP = IPS e.maxZirCAD CEREC/in Lab MT Multi, CE = CeramillZolidfx ML c = cervical, m = medium, i = incisal.

**Figure 6 materials-15-03606-f006:**
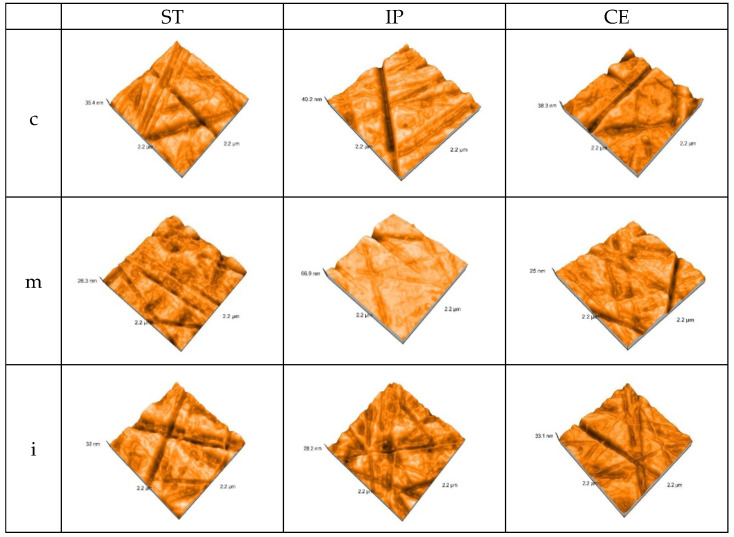
3D AFM images of the polished samples, after aging.ST= STML, IP= IPS e.maxZirCAD CEREC/in Lab MT Multi, CE = CeramillZolidfx ML, c = cervical, m = medium, i = incisal.

**Figure 7 materials-15-03606-f007:**
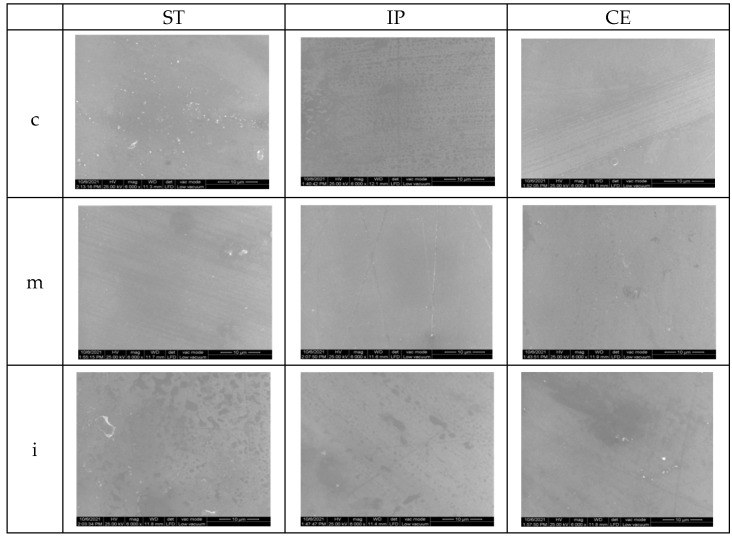
SEM images for glazed samples before aging. ST= STML, IP= IPS e.maxZirCAD CEREC/in Lab MT Multi, CE= CeramillZolidfx ML, c = cervical, m = medium, i = incisal.

**Figure 8 materials-15-03606-f008:**
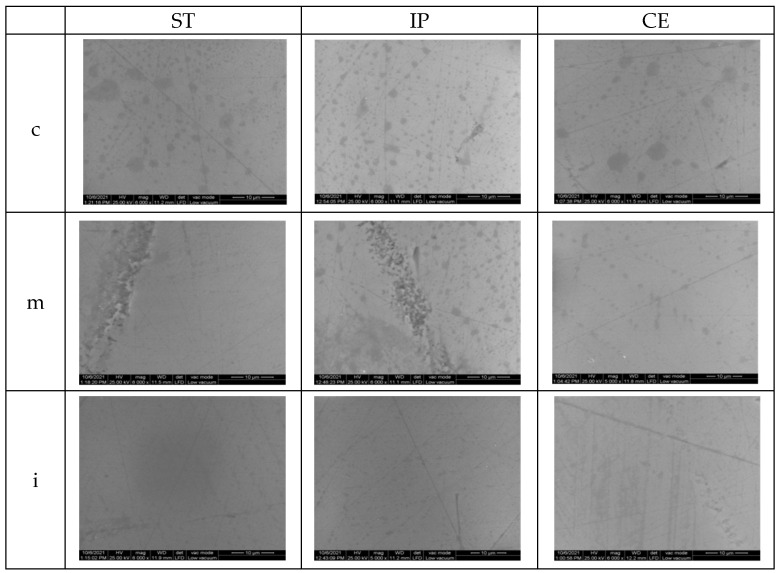
SEM images for polished samples before aging.ST= STML, IP= IPS e.maxZirCAD CEREC/in Lab MT Multi, CE= CeramillZolidfx ML, c = cervical, m = medium, i = incisal.

**Figure 9 materials-15-03606-f009:**
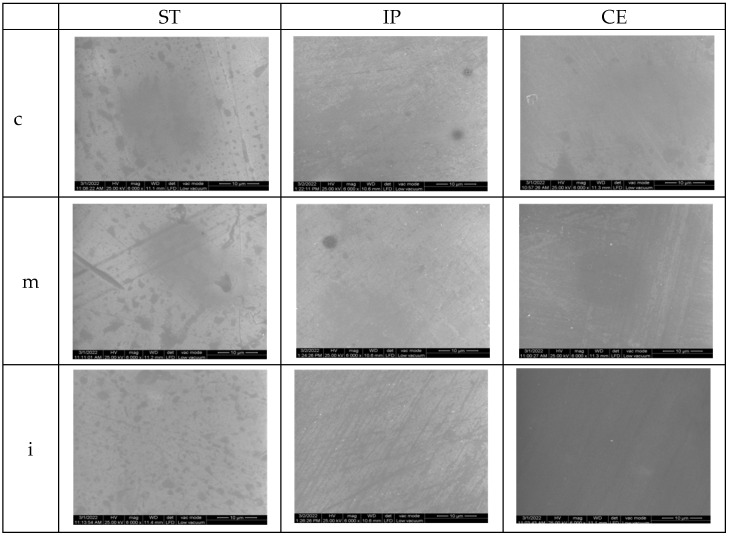
SEM images for glazed samples after aging.ST = STML, IP = IPS e.maxZirCAD CEREC/in Lab MT Multi, CE = CeramillZolidfx ML, c = cervical, m = medium, i = incisal.

**Figure 10 materials-15-03606-f010:**
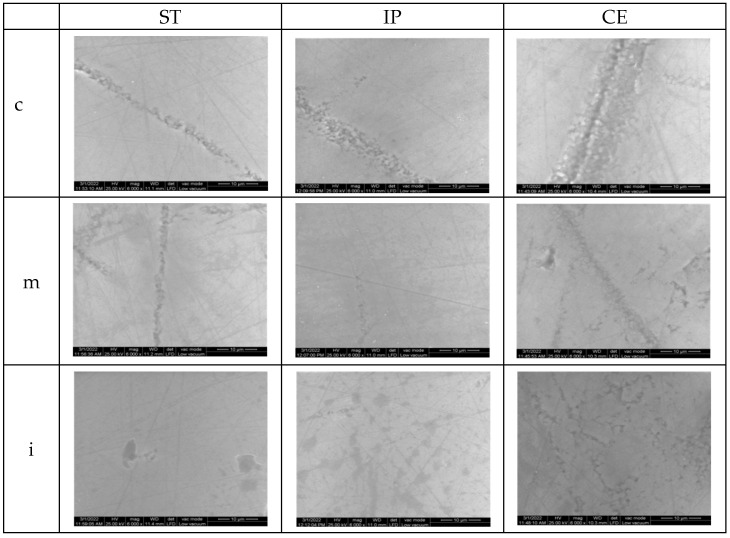
SEM images for polished samples after aging.ST = STML, IP = IPS e.maxZirCAD CEREC/in Lab MT Multi, CE = CeramillZolidfx ML, c = cervical, m = medium, i = incisal.

**Figure 11 materials-15-03606-f011:**
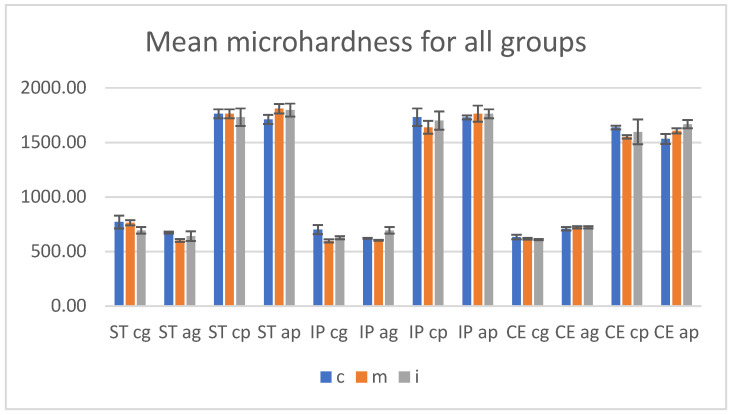
Mean microhardness values and SD before and after aging. ST = STML, IP = IPS e.maxZirCAD CEREC/in Lab MT Multi, CE = CeramillZolidfx ML; cg = control glazed, cp = control polished, ag = aged glazed, ap = aged polished;c = cervical, m = medium, i = incisal.

**Table 1 materials-15-03606-t001:** Properties of the materials taken into the study [49,50,51].

Restauration Material	Manufacturer	Translucency/ Shade	Yttrium Content	Flexural Strength
STML (ST)	Katana, Kuraray Noritake Dental, Japan	Super translucency A2	4 mol%	750 MPa
IPS e.max Zir CAD (IP)	Ivoclar Vivadent AG, Schaan, Liechtenstein	Super-high translucency + Super translucency A2	4 mol%-incisal 4 + 5 mol%-transition 4 mol%-cervical	850 MPa
Ceramill Zolid fx ML (CE)	Amman Girrbach, AG, Koblach, Austria	Super-high translucency A2/A3	5 mol%	700 MPa

**Table 2 materials-15-03606-t002:** Mean values with SD for Ra, Rz parameters of the samples, before aging (n = 8).

		ST cg	ST cp	IP cg	IP cp	CE cg	CE cp
Ra	c	0.064 ± 0.009	0.041 ± 0.009	0.073 ± 0.009	0.043 ± 0.010	0.086 ± 0.010	0.047 ± 0.009
	m	0.065 ± 0.009	0.043 ± 0.010	0.080 ± 0.009	0.044 ± 0.009	0.085 ± 0.010	0.045 ± 0.010
	i	0.067 ± 0.010	0.042 ± 0.010	0.083 ± 0.009	0.046 ± 0.009	0.084 ± 0.008	0.047 ± 0.010
Rz	c	0.445 ± 0.099	0.360 ± 0.094	0.475 ± 0.098	0.366 ± 0.098	0.580 ± 0.096	0.386 ± 0.082
	m	0.466 ± 0.096	0.363 ± 0.091	0.530 ± 0.091	0.371 ± 0.096	0.573 ± 0.098	0.371 ± 0.091
	i	0.493 ± 0.093	0.378 ± 0.100	0.533 ± 0.100	0.385 ± 0.092	0.563 ± 0.090	0.373 ± 0.093

Ra (µm)—arithmetic average surface roughness, Rz (µm)—maximum surface roughness. ST= STML, IP= IPS e.maxZirCAD CEREC/in Lab MT Multi, CE= CeramillZolidfx ML. cg = control glazed, cp = control polished, c = cervical, m = medium, i = incisal.

**Table 3 materials-15-03606-t003:** Mean values with SD for Ra, Rz parameters of the samples, after aging (n = 8).

		ST ag	ST ap	IP ag	IP ap	CE ag	CE ap
Ra	c	0.080 ± 0.009	0.054 ± 0.010	0.081 ± 0.009	0.052 ± 0.009	0.088 ± 0.009	0.048 ± 0.008
	m	0.078 ± 0.009	0.055 ± 0.010	0.084 ± 0.008	0.050 ± 0.009	0.087 ± 0.009	0.047 ± 0.009
	i	0.077 ± 0.009	0.052 ± 0.010	0.086 ± 0.010	0.049 ± 0.008	0.086 ± 0.009	0.048 ± 0.009
Rz	c	0.535 ± 0.100	0.426 ± 0.101	0.543 ± 0.094	0.372 ± 0.090	0.576 ± 0.099	0.413 ± 0.095
	m	0.520 ± 0.095	0.422 ± 0.099	0.548 ± 0.097	0.383 ± 0.091	0.579 ± 0.095	0.406 ± 0.091
	i	0.518 ± 0.092	0.410 ± 0.091	0.562 ± 0.089	0.390 ± 0.094	0.591 ± 0.084	0.416 ± 0.090

Ra-arithmetic average surface roughness, Rz-maximum surface roughness. ST= STML, IP= IPS e.maxZirCAD CEREC/in Lab MT Multi, CE= CeramillZolidfx ML. ag = aged glazed, ap = aged polished, c = cervical, m = medium, i = incisal.

**Table 4 materials-15-03606-t004:** The unpaired *t*-Test test (α = 0.05)-*p* values regarding Ra, Rz for control, glazed-control polished, respectively aged glazed-aged polished groups in each area.

		ST cg-cp	IP cg-cp	CE cg-cp	ST ag-ap	IP ag-ap	CE ag-ap
Ra	c	0.001	<0.001	<0.001	0.002	<0.001	<0.001
	m	0.004	<0.001	<0.001	<0.001	<0.001	<0.001
	i	<0.001	<0.001	<0.001	<0.001	<0.001	<0.001
Rz	c	0.039	0.026	0.004	0.042	0.005	0.006
	m	0.044	0.009	0.006	0.039	0.005	0.003
	i	0.031	0.005	0.002	0.046	0.003	0.001

Ra-arithmetic average surface roughness, Rz-maximum surface roughness. ST = STML, IP = IPS e.maxZirCAD CEREC/in Lab MT Multi, CE = CeramillZolidfx ML. cg = control glazed, cp = control polished, ag = aged glazed, ap = aged polished. c = cervical, m = medium, i = incisal.

**Table 5 materials-15-03606-t005:** The paired *t*-Test test (α = 0.05)-*p* values regarding Ra, Rz for control glazed–agedglazed, respectively control polished–aged polished groups in each area.

		ST cg-ag	ST cp-ap	IP cg-ag	IP cp-ap	CE cg-ag	CE cp-ap
Ra	c	0.006	0.049	0.134	0.058	0.715	0.868
	m	0.046	0.022	0.521	0.387	0.662	0.628
	i	0.042	0.047	0.583	0.442	0.311	0.808
Rz	c	0.371	0.225	0.775	0.621	0.583	0.866
	m	0.412	0.425	0.621	0.539	0.715	0.754
	i	0.344	0.568	0.592	0.546	0.662	0.862

Ra-arithmetic average surface roughness, Rz-maximum surface roughness. ST = STML, IP = IPS e.maxZir CAD CEREC/in Lab MT Multi, CE = CeramillZolidfx ML. cg = control glazed, cp = control polished, ag = aged glazed, ap=aged polished. c = cervical, m = medium, i = incisal.

**Table 6 materials-15-03606-t006:** Sa, Sq values for control-glazed and control-polished samples, in each area (n = 8).

		ST cg	IP cg	CE cg	ST cp	IP cp	CE cp
Sa	c	1.651	1.960	1.723	1.048	1.244	1.388
	m	2.424	2.536	2.662	1.315	1.381	1.582
	i	1.524	1.598	1.924	1.529	1.529	1.343
Sq	c	1.934	2.293	1.981	1.348	1.335	1.738
	m	2.945	3.049	3.123	1.691	1.552	2.113
	i	1.842	1.996	2.144	1.945	2.045	1.545

Sa = arithmetical mean height, Sq = root mean square height. ST = STML, IP = IPS e.maxZirCAD CEREC/in Lab MT Multi, CE = CeramillZolidfx ML. cg = control glazed, cp = control polished, c = cervical, m = medium, i = incisal.

**Table 7 materials-15-03606-t007:** Sa, Sq values for aged-glazed and aged-polished samples, in each area (n = 8).

		ST ag	IP ag	CE ag	ST ap	IP ap	CE ap
Sa	c	23.334	28.134	31.817	20.651	27.714	20.342
	m	21.419	29.050	27.315	31.783	27.674	22.991
	i	22.458	24.262	24.464	28.978	25.502	20.515
Sq	c	26.744	32.526	36.371	23.669	31.778	23.529
	m	24.414	33.221	31.242	36.254	31.952	25.883
	i	25.702	27.911	27.873	32.988	29.106	23.728

**Table 8 materials-15-03606-t008:** Mean Vickers hardness values with SD, for all samples (n = 8).

		ST Glazed	ST Polished	IP Glazed	IP Polished	CE Glazed	CE Polished
control groups	c	770.33 ±9.60	1763 ± 17.02	700 ± 5.72	1732 ± 15.70	632.67± 8.50	1637 ± 12.50
	m	763.67 ± 8.62	1763 ± 17.02	597 ± 3.80	1638.67 ± 14.02	617 ± 7.90	1552 ± 8.41
	i	694.67 ± 9.14	1732 ± 15.89	626 ± 4.20	1700.67 ± 15.03	610 ± 7.28	1597 ± 9.54
aged groups	c	672.67 ± 8.32	1711 ± 15.41	621 ± 4.10	1729.67 ± 15.05	708.33 ± 9.42	1532.33 ± 9.50
	m	601 ± 5.12	1809.67 ± 17.91	603.33 ± 4.08	1764.33 ± 16.15	722± 8.11	1607.67 ± 8.97
	i	640.33 ± 5.81	1797 ± 17.56	694.33 ± 4.21	1762.67 ± 16.08	722± 8.11	1667.67± 5.70

ST = STML, IP = IPS e.maxZirCAD CEREC/in Lab MT Multi, CE = CeramillZolidfx ML; c = cervical, m = medium, i = incisal.

**Table 9 materials-15-03606-t009:** The paired *t*-Test test (α = 0.05)-*p* values for microhardnessamong control glazed-aged glazed, respectively control polished-aged polished groups in each area.

VN	ST cg-ag	STcp-ap	IP cg-ag	IP cp-ap	CE cg-ag	CE cp-ap
c	0.621	0.033	0.887	0.972	0.012	0.091
m	0.116	0.248	0.584	0.623	0.016	0.183
i	0.413	0.475	0.671	0.523	0.001	0.359

VN = Vickers Hardness number (microhardness value). ST = STML, IP = IPS e.maxZirCAD CEREC/in Lab MT Multi, CE = CeramillZolidfx ML. cg = control glazed, cp=control polished, ag = aged glazed, ap=aged polished. c = cervical, m = medium, i = incisal.

## Data Availability

Not applicable.

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
