# Peer review of "Surface Characteristics of High Translucent Multilayered Dental Zirconia Related to Aging"

_materials, 2022, doi:10.3390/ma15103606_

Round 1
Reviewer 1 Report
ORIGINALITY: The originality of the paper Surface characteristics of high translucent multi-layered dental zirconia related to aging is good. In fact, I went through the current literature on the matter, and I found only a few recent and well written articles on similar topics.
The authors need to perform the following major changes:
- Authors have pushed their work on the role of mechanical and chemical factors, but no comparation to other aspects was reported in the introductory section; moreover, clinical applications are poorly reported about this topic, and some concerns about nanoparticles releasing should be reported and briefly discussed as well (see and discuss: Bressan, E., Ferroni, L., Gardin, C., Bellin, G., Sbricoli, L., Sivolella, S., Brunello, G., Schwartz-Arad, D., Mijiritsky, E., Penarrocha, M., Penarrocha, D., Taccioli, C., Tatullo, M., Piattelli, A., & Zavan, B. (2019). Metal Nanoparticles Released from Dental Implant Surfaces: Potential Contribution to Chronic Inflammation and Peri-Implant Bone Loss. Materials (Basel, Switzerland), 12(12), 2036.)
- Main limitations should be reported
- Future prospectives must be related to clinical strategies: please improve/add this concept accordingly.
- Please explain all the acronyms throughout the text
- Authors may also compare such materials and approach with other promising biomaterials, such as the biomedical applications of several new biomaterials like the Phosphorene or the Borophene. (Phosphorene Is the New Graphene in Biomedical Applications. Materials (Basel, Switzerland), 12(14), 2301. – AND - Borophene Is a Promising 2D Allotropic Material for Biomedical Devices. Appl. Sci. 2019, 9, 3446. )
Minor:
Some typos are present here and there: please, check and fix the text.
Author Response
- Authors have pushed their work on the role of mechanical and chemical factors, but no comparation to other aspects was reported in the introductory section; moreover, clinical applications are poorly reported about this topic, and some concerns about nanoparticles releasing should be reported and briefly discussed as well (see and discuss: Bressan, E., Ferroni, L., Gardin, C., Bellin, G., Sbricoli, L., Sivolella, S., Brunello, G., Schwartz-Arad, D., Mijiritsky, E., Penarrocha, M., Penarrocha, D., Taccioli, C., Tatullo, M., Piattelli, A., &Zavan, B. (2019). Metal Nanoparticles Released from Dental Implant Surfaces: Potential Contribution to Chronic Inflammation and Peri-Implant Bone Loss. Materials (Basel, Switzerland), 12(12), 2036.)
Response: We agree to this point. We included clinical applications for the tested materials studied in this manuscript. We included the reference that you sent in the manuscript- [11-13].
- Main limitations should be reported
Response: We agree to this point and included the following informationregarding the main limitations of the materials and of the study.
“For super-high translucent 5Y-TZP (third-generation), the Yâ‚‚O₃ content was increased to 5 mol% and the quantity of the cubic phase rises up to 50%, also the amount of tetragonal phase decreases, leading to a lower possibility of tetragonal to monoclinic phase transformation, with reduction in strength and fracture toughness [8]; that is a disadvantage of these new translucent materials and the clinicians should use the 4Y and 5Y zirconia when the aesthetic needs are required.”
“It was observed that the surface roughness and the microstructure characteristics can affect the strength of the material [22] and be relevant factors that influence the appearance, the final color or bacterial accumulation”. [23,24]
‘’The limitation of this study was the short-term aging and the fact that certain factors that interact with restoration in oral environment could not be fully reproduced (chewing force, saliva components, temperature and ph fluctuations).’’
- Future prospective must be related to clinical strategies: please improve/add this concept accordingly.
Response: We agree to this point and added additional information related to the future prospective.
“The strategy for improving the optical performance of material imply microstructural variations - the higher Yttrium content influence the coefficient of thermal expansion, the size of grains (they are larger), reduce the residual porosity, resulting fewer boundaries, less birefringence and scattering of the light [8] leading to higher translucency [9]. The nanoparticles zirconia is presumed to be the next progress in evolution of this material”. [11-13].
“In order to obtain materials with a high translucency, but also resistant, new grades of zirconia have been developed, pre-sintered blocks being available as pre-shaded in various tonalities or multilayered with gradients of chroma.” [2]
- Please explain all the acronyms throughout the text
Response: We agree to this point and corrected accordingly.
- Authors may also compare such materials and approach with other promising biomaterials, such as the biomedical applications of several new biomaterials like the Phosphorene or the Borophene. (Phosphorene Is the New Graphene in Biomedical Applications. Materials (Basel, Switzerland), 12(14), 2301. – AND - Borophene Is a Promising 2D Allotropic Material for Biomedical Devices. Appl. Sci. 2019, 9, 3446.)
Response: We agree to this point and added this reference in the manuscript- [11-13].
Minor:’
Some typos are present here and there: please, check and fix the text.
Response: We corrected accordingly.

Reviewer 2 Report
This review is primarily a statistical one, with an overarching recommendation and specific major points. My comments to the authors are as follows:
- Unfortunately, this manuscript as a whole does not meet accepted quality standards for a research article. The presentation of the results is fragmented and too much is packed into one manuscript.
- The study design of this study was a laboratory work which should be clearly stated in the abstract and methods section.
- The quality of statistical reporting was poor: I scored 2 in a scale from 0 (poor) to 10 (very high). You should improve the quality of statistical reporting and data presentation Please see comments below.
- My interpretation is that there are several response variables, two grouping variables (materials and glazed/polished/control/autoclaved). These form 12 independent groups, n=8 in each sub-group. In addition, there are three repeated measurements (c, m, i). Please clarify this for your readers in the Statistical analysis sub-section.
- Help your readers and provide more information about the data analysis. Within the Statistical analysis section, you need to explain which statistical methods or tests were used for each analysis, rather than just listing all the statistical methods used in one place. In a well-written statistical analysis subsection, authors should also identify the variables used in each analysis. Readers should be informed which variables were analysed with each method. Please define your main outcome variables and explanatory factors.
- Please describe in the Statistical analysis subsection did the data satisfactorily fulfil the underlying assumptions and preconditions of the main analysis methods.
- When evaluating the validity of the findings, the reader must know the number of study participants. Sample size is an important consideration for research. The total number of participants or sample size of each group should be clearly reported in all tables and figures.
- Tables and figures should be able to stand alone. That is, all information necessary for interpretation should be included within the table or figure, legend, or footnotes. This means that descriptive statistics, significance tests and multivariable modelling methods used are named. All abbreviations should be defined in each table and figure, in the title or as a footnote. Even if they are described in the method section. Please revise tables and figures.
- The presentation of the results is difficult to follow. There are many tables, and the analyses are presented in a stratified manner. Significance tests are abundant. A two-way analysis of variance (material and glazed/polished/control/autoclaved as grouping factors) would be more appropriate for analysing the data. In addition, I recommend using multivariable linear regression with GEE extension or mixed linear models to handle the repeated measures.
- Well-prepared figures could help in the data presentation.
Author Response
- Unfortunately, this manuscript as a whole does not meet accepted quality standards for a research article. The presentation of the results is fragmented and too much is packed into one manuscript.
Response: We agree to this point and removed from the manuscript the unnecessary pieces of information and improved the form and the structure of the manuscript to be able to meet the standards for a research article.
- The study design of this study was a laboratory work which should be clearly stated in the abstract and methods section.
Response: We agree to this point and added information related to the laboratory work in the abstract and in the method section.
- The quality of statistical reporting was poor: I scored 2 in a scale from 0 (poor) to 10 (very high). You should improve the quality of statistical reporting and data presentation Please see comments below.
- My interpretation is that there are several response variables, two grouping variables (materials and glazed/polished/control/autoclaved). These form 12 independent groups, n=8 in each sub-group. In addition, there are three repeated measurements (c, m, i). Please clarify this for your readers in the Statistical analysis sub-section.
Response: We agree to this point. One-way ANOVA and t-test were used as statistical analysis. We added in the results and discussion sections new information to clarify in the statistical section.
- Help your readers and provide more information about the data analysis. Within the Statistical analysis section, you need to explain which statistical methods or tests were used for each analysis, rather than just listing all the statistical methods used in one place. In a well-written statistical analysis subsection, authors should also identify the variables used in each analysis. Readers should be informed which variables were analyzed with each method. Please define your main outcome variables and explanatory factors.
Response: We agree to this point and added in the manuscript pieces of information.
“One way Anova test was used for statistical analysis of variables recorded on glazed and polished sides, for comparing areas on a surface or between each area of different treated surfaces - in one stage of the study- before or after aging. T-test was used to analyze the variables of the two stages – before and after aging, for a material and the significant differences as a result of autoclaving.
For comparison of surface roughness between the sample sets, a significance level of α =0.05 was established. Correlation between Ra, Rz and microhardness was estimated using Pearson’s correlations.”
- Please describe in the Statistical analysis subsection did the data satisfactorily fulfil the underlying assumptions and preconditions of the main analysis methods.
Response: We agree to this point. The sample size was selected based on the confidence level and the margin of error. The power of the statistical test was calculated with the same software IBM SPSS Statistics. The power test was determined before the sample size to ensure that the sample size is enough for the aim of this research.
- When evaluating the validity of the findings, the reader must know the number of study participants. Sample size is an important consideration for research. The total number of participants or sample size of each group should be clearly reported in all tables and figures.
Response:Sample size was calculated using a specific software (G*Power software 3.1.9.4 (University Kiel, Kiel, Germany)). The effect size was chosen 0,50. The calculation of the test revealed that a number of 8 samples for each groupis enough. This is why this research included 8 samples for each studied group.
- Tables and figures should be able to stand alone. That is, all information necessary for interpretation should be included within the table or figure, legend, or footnotes. This means that descriptive statistics, significance tests and multivariable modelling methods used are named. All abbreviations should be defined in each table and figure, in the title or as a footnote. Even if they are described in the method section. Please revise tables and figures.
Response: We agree to this point and added a new table in the manuscript regarding the statistical analysis. We added as well abbreviations in each table and figure. We revised the tables and figures.
- The presentation of the results is difficult to follow. There are many tables, and the analyses are presented in a stratified manner. Significance tests are abundant. A two-way analysis of variance (material and glazed/polished/control/autoclaved as grouping factors) would be more appropriate for analysing the data. In addition, I recommend using multivariable linear regression with GEE extension or mixed linear models to handle the repeated measures.
Response:We used one-way ANOVA and the t-test for the samples where we had only one variable, for example, the material type.
- Well-prepared figures could help in the data presentation.
Response: We agree to this point and improved the figures in the presentation.

Reviewer 3 Report
The paper is interesting and can be a contribution to the scientific field.
However, I suggest some changes in order to improve the overall quality of the manuscript for the readers.
Lines: 150-156:
Please add the cut-off values used with the SJ-201
Please outline better (in a specific paragraph) the role of autoclave aging. It leads to hydrothermal degradation causing an increase in the cubic phase volume and a separation of cubic and tetragonal phases.
The authors could add a sentence before the conclusions that could outline the limits of some of the investigated materials.
For example a sentece like the following could improve the discussion and the data relative to ST:
Around line 583 or around lines 440 where differences in mechanical properties are outlined.
“The lowest surface characteristics of ST is also confirmed by other papers investigating other mechanical properties. External gap progression of ST overlays and crowns is significantly higher in respect to lithium silicate ones (https://doi.org/10.1111/jerd.12837). This finding is supported by Reyes et al that reported that ST presented the lowest flexural strength among translucent and high-strength zirconia (https://doi.org/10.1016/j.prosdent.2021.06.019)”
Author Response
Lines: 150-156:
Please add the cut-off values used with the SJ-201
Response: We agree to this point and added the information regarding to cut-off value.
“cut-off value was 0.3 mm”
Please outline better (in a specific paragraph) the role of autoclave aging. It leads to hydrothermal degradation causing an increase in the cubic phase volume and a separation of cubic and tetragonal phases.
Response: We agree to this point and included additional information regarding to role of autoclave aging.
“Negative effects were reported after simulated accelerating aging of zirconia - LTD (low temperature degradation), by autoclaving”[17]
“The aging process is associated with a slow tetragonal to monoclinic phase transformation, 4% volume increase of particles, micro-cracks with water penetration, grain detachment, decrease in hardness and fracture strength, surface roughening and degradation at the surface (and subsurface) which is in contact with moisture” [1,18,19].
The authors could add a sentence before the conclusions that could outline the limits of some of the investigated materials.
For example, a sentence like the following could improve the discussion and the data relative to ST:
Around line 583 or around lines 440 where differences in mechanical properties are outlined.
“The lowest surface characteristics of ST is also confirmed by other papers investigating other mechanical properties. External gap progression of ST overlays and crowns is significantly higher in respect to lithium silicate ones (https://doi.org/10.1111/jerd.12837). This finding is supported by Reyes et al that reported that ST presented the lowest flexural strength among translucent and high-strength zirconia (https://doi.org/10.1016/j.prosdent.2021.06.019)”
Response: We agree to this point and included the following information regarding the main limitations of the materials and of the study.
This means that the material with 4mol % Yttrium content is more susceptible to the aging process, thus hydrothermal and mechanical impact cause phase transformation, resulting in surface degradation and an increase in microhardness-this situation has been presented in other studies [57,58].

Round 2
Reviewer 2 Report
I noticed that you have not made the necessary corrections to the statistical analyses. The main issues are as follows:
- Please apply two-way analysis of variance with RA and Rz as response variables and materials (ST,IP and CE) and handling (polishing vs glazing) as grouping factors, i.e. you have two grouping factors. These analyses should be performed separately for before and after study units.
- Note that cervical, medium, and incisal areas are not independent groups. When comparing these groups, you should apply a statistical method for repeated measures (e.g. analysis of variance with repeated measures).
- Using analysis of variance requires that the outcome variable is normally distributed. Have you checked this assumption?
- Report sample sizes in all tables and figures.
- Pleased do not refer to variables or measurements by the term "parameters".
- Consulting a biostatistician could help to solve the main statistical issues.
Author Response
- Please apply two-way analysis of variance with RA and Rz as response variables and materials (ST, IP and CE) and handling (polishing vs glazing) as grouping factors, i.e. you have two grouping factors. These analyses should be performed separately for before and after study units.
Response: We agree to this point. We used two-way ANOVA, unpaired t-Test and paired t-Test for statistical analysis. We added in Results and Discussion section new information to clarify in the statistical section.
- Note that cervical, medium, and incisal areas are not independent groups. When comparing these groups, you should apply a statistical method for repeated measures (e.g. analysis of variance with repeated measures).
Response: We agree to this point.
The cervical, medium and incisal areas were considered to be different groups because they are obtained from different formulas in two out of three cases. But for a general overview we run comparative tests in both scenarios: when the groups are independent and when the groups are dependent. After running the statistical analysis, we obtained insignificant differences (p>0.05) in all cases.
- Using analysis of variance requires that the outcome variable is normally distributed. Have you checked this assumption?
Response: We agree to this point and added information in the statistical section
“To decide upon the statistical type of tests we applied the Shapiro-Wilk test for distribution and obtained that the data are normally distributed (p>0.05) in most of the cases”
- Report sample sizes in all tables and figures.
Response: We agree to this point and added the sample size in tables and figures: (n=8)
- Pleased do not refer to variables or measurements by the term "parameters".
Response: We agree to this point.
- Consulting a biostatistician could help to solve the main statistical issues.
Response: Thank you for your suggestion. I consulted a statistician and I reviewed my entire statistics. The contact person for statistical expertise is Associate Professor MOLERIU LAVINIA CRISTINA, PhD in Mathematics and Computer Science from Department of Functional Science at “Victor Babes” University of Medicine and Pharmacy Timisoara, email: moleriu.lavinia@umft.ro
